# MHC-mismatched synovial mesenchymal stem cell injections delay knee osteoarthritis progression through hepatocyte growth factor secretion in rats

Tsukasa Kitahashi[1], Kentaro Nakamura[1], Ryo Kogawa[1], Ichiro Sekiya [2]*

1 Bioscience & Engineering Laboratory, FUJIFILM Corporation, Kanagawa, Japan, 2 Center for Stem Cell and Regenerative Medicine, Institute of Science Tokyo, Tokyo, Japan

* sekiya.arm@tmd.ac.jp

## Abstract

### Background

Knee osteoarthritis (OA) currently lacks disease-modifying treatments. While autologous mesenchymal stem cell (MSC) transplantation has shown promise, its limitations have led to growing interest in allogeneic MSC therapy. The aim of this study was to clarify the functional role of hepatocyte growth factor (HGF) in the chondroprotective effects of synovial MSCs transplanted under fully major histocompatibility complex (MHC)-mismatched allogeneic conditions in a rat model.

### Methods

Synovial MSCs were isolated from the knees of ACI rats and transfected with either *Hgf* siRNA or control siRNA. HGF knockdown was confirmed at both the mRNA and protein levels. Chondrogenic capacity was assessed by evaluating the cartilage pellet size after in vitro differentiation. OA was surgically induced in Lewis rats by anterior cruciate ligament transection (ACLT). At one week post-surgery, the rats (n = 20 knees/group) received weekly intra-articular injections of one of the following for 7 weeks: (1) vehicle control (medium alone), (2) control siRNA-treated MSCs, or (3) *Hgf* siRNA-treated MSCs (1 × 10⁶ cells). At 8 weeks post-injection, cartilage degeneration was evaluated using macroscopic and histological Osteoarthritis Research Society International (OARSI) scoring.

### Results

Transfection with *Hgf* siRNA reduced *Hgf* gene expression to 30% of the control levels and significantly decreased HGF protein secretion. HGF knockdown impaired in vitro chondrogenic differentiation, resulting in a reduced cartilage pellet size. In vivo, treatment with control siRNA-treated MSCs significantly delayed OA progression

which permits unrestricted use, distribution, and reproduction in any medium, provided the original author and source are credited.

**Data availability statement:** All relevant data are within the manuscript and its Supporting information files.

**Funding:** This study was funded by Fujifilm Corporation. The funder provided research funding to IS, and support in the form of salaries for authors TK, KN, and RK. According to the author contributions, the funder-affiliated authors (TK, KN, and RK) participated in conceptualization, methodology, investigation, resources, data curation, formal analysis, project administration, and validation. The funder did not have any role in the decision to publish or the preparation of the manuscript.

**Competing interests:** Tsukasa Kitahashi, Kentaro Nakamura, and Ryo Kogawa are employees of Fujifilm Corporation. The remaining authors declare no competing interests. This does not alter our adherence to PLOS ONE policies on sharing data and materials.

**Abbreviations:** OA, Osteoarthritis; MSC, Mesenchymal stem cell; HGF, Hepatocyte growth factor; siRNA, Small interfering RNA; ELISA, Enzyme-Linked Immunosorbent Assay; ACLT, Anterior cruciate ligament transection; OARSI, Osteoarthritis Research Society International.

compared to the vehicle group, while *Hgf* siRNA-treated MSCs demonstrated diminished therapeutic efficacy, indicating reduced chondroprotective effects when HGF secretion was suppressed.

## Conclusions

Synovial MSCs derived from ACI rats effectively delayed OA progression in Lewis rats despite the full MHC mismatch. HGF secretion played a critical role in mediating these chondroprotective effects. These findings highlight HGF as a key therapeutic effector in MSC-based treatment of OA under immunologically mismatched allogeneic conditions.

## Introduction

Osteoarthritis (OA) is a degenerative joint disease characterized by cartilage degradation, synovial inflammation, and structural changes in the subchondral bone that ultimately lead to joint dysfunction and chronic pain [1]. OA progression is driven by multiple factors, including cartilage matrix breakdown, chondrocyte apoptosis, and elevated levels of inflammatory cytokines [2]. Due to this multifactorial pathophysiology, therapeutic approaches that can target several pathways simultaneously—such as cell-based therapies—are attracting growing interest.

Synovial mesenchymal stem cells (MSCs) have shown promising chondroprotective effects when administered intra-articularly, as demonstrated in both preclinical [3] and clinical studies [4]. We previously reported that intra-articular injection of autologous synovial MSCs can suppress OA progression in humans [4]. However, autologous transplantation presents several challenges, including donor site morbidity, extended culture periods, and high manufacturing costs. These limitations have prompted increasing interest in allogeneic MSC therapy, which offers advantages in terms of availability, standardization, and scalability.

A key concern arising during allogeneic MSC therapy is immunological incompatibility between the donor and recipient, particularly regarding major histocompatibility complex (MHC) antigens. Previous studies using rat meniscectomy models have shown that meniscal regeneration is poorer following transplantation of fully MHC-mismatched MSCs than following transplantation of syngeneic or partially matched MSCs [5,6]. This observation suggests that immune rejection can negatively impact therapeutic outcomes; however, the efficacy of using fully mismatched MSCs in OA models remains poorly understood.

Given the limited understanding of the underlying mechanisms controlling the therapeutic responses observed following the transplantation of MHC-mismatched MSCs in OA, clarifying the roles of paracrine factors secreted by MSCs becomes essential. Among the various paracrine factors, hepatocyte growth factor (HGF) is of particular interest due to its well-documented anti-inflammatory, antifibrotic, and tissue-regenerative properties [7]. HGF stimulates chondrocyte proliferation, enhances matrix production, and suppresses catabolic cytokines, and the benefits of

the administration of exogenous HGF or HGF-overexpressing MSCs have been reported [8,9]. Nevertheless, the functional importance of endogenously secreted HGF—especially under MHC-mismatched allogeneic conditions—remains unclear. However, the possibility remains that HGF may serve as a critical mediator of MSC therapeutic efficacy, even in immunologically challenging environments.

The aim of the present study was to clarify the functional role of HGF in establishing the chondroprotective effects of synovial MSCs under fully MHC-mismatched allogeneic conditions. We hypothesized that HGF serves as a key mediator of MSC efficacy, even under immunologically challenging conditions. To test this hypothesis, we employed a rat OA model using ACI rats (donors) and Lewis rats (recipients), which differ markedly in MHC class I and II antigens and are known to elicit robust immune responses [5]. Synovial MSCs were transfected with Hgf siRNA and injected intra-articularly, and the effects of HGF suppression on both in vivo chondroprotective efficacy and in vitro chondrogenic capacity were evaluated. Importantly, our study specifically focuses on OA cartilage rather than meniscal tissue, and employs HGF knockdown to mechanistically dissect the role of this specific paracrine factor under stringent immunological conditions—an approach not previously undertaken in fully MHC-mismatched settings.

## Materials and methods

### Animals

This study was conducted in accordance with the ARRIVE 2.0 guidelines. All animal procedures were approved by the Institutional Animal Care and Use Committee of FUJIFILM Corporation, with the following details: (1) Title of the approved project: *Establishment and Evaluation of HGF Knockdown Allogeneic Synovial Stem Cells from ACI Rats in an ACLT Rat Osteoarthritis Model*; (2) Name of the ethics committee: Institutional Animal Care and Use Committee of FUJIFILM Corporation; (3) Approval numbers: A-1–210092 and A-1–220037; (4) Dates of approval: December 12, 2021 and September 8, 2022.

To establish MHC-mismatched synovial MSCs, 30 four-week-old male ACI rats (ACI/N Slc), weighing 50–70 g, were purchased from Japan SLC, Inc. (Hamamatsu, Japan). Fifteen of these rats were used for pooled synovial MSC preparation for transplantation, and the remaining 15 were used to confirm the tissue harvesting protocol.

To evaluate regenerative efficacy and underlying molecular mechanisms, 34 female Lewis rats (LEW/CrlCrlj), aged 9–10 weeks and weighing 150–170 g, were obtained from Jackson Laboratory Japan, Inc. (Yokohama, Japan). Four of these animals were procured as potential replacements for contingency use.

All animals were housed under standard laboratory conditions with ad libitum access to food and water. Environmental conditions were maintained at a temperature of 20–26°C, relative humidity of 30–70%, and a 12 h light/dark cycle.

At 8 weeks after the first intra-articular injection, all animals were euthanized via exsanguination through abdominal aortic transection under general anesthesia with 2.5% isoflurane administered via face mask. Throughout the study, all efforts were made to minimize animal distress and suffering. This included appropriate anesthesia, careful handling, postoperative pain management, and routine monitoring of animal health and behavior. All procedures were performed by trained personnel in accordance with institutional guidelines for humane animal care.

### Isolation of rat synovial MSC

Rat synovial MSCs were isolated as previously described [10]. After an adaptation period of at least 7 days, male ACI rats were euthanized by exsanguination from the inferior vena cava under isoflurane anesthesia. Synovial tissue harvested from the infrapatellar fat pad of both knees from 15 rats was pooled and minced, followed by digestion with Liberase MNP-S (Roche Diagnostics Corp., IN, USA) in a water bath at 37°C for 2 h. The resulting isolated cells were cultured in α-minimum essential medium (αMEM; FUJIFILM Wako Chemicals, Tokyo, Japan) supplemented with 20% fetal bovine serum (FBS; Thermo Fisher Scientific, MA, USA) for 8 days at 37°C in 5% $CO_2$. The cells were then harvested and

cryopreserved as original stocks in CP-1 High Grade cryoprotectant (Kyokuto Pharmaceutical Industrial Co. Ltd., Tokyo, Japan) at −150°C (CLN-1700CWE, Nihon Freezer, Tokyo, Japan) (Passage 0). These isolation procedures were performed in three separate sessions. For experiments, synovial MSCs from the original stocks (Passage 0) were cultured for 7 days, harvested, and cryopreserved in CP-1 High Grade cryoprotectant (Passage 1). The cells were used for subsequent analyses without any sorting [6]. For cryopreservation, CP-1 High Grade was supplemented with 25% human recombinant albumin (Albumin, Human, recombinant expressed in plants; FUJIFILM Wako Pure Chemical Corp.). The cell suspension, prepared in physiological saline, was subsequently combined with an equal volume of the CP-1 albumin solution, resulting in a final composition of 6% hydroxyethyl starch, 5% dimethyl sulfoxide, and 4% human recombinant albumin.

### Transfection of synovial MSCs with *Hgf* siRNA

Synovial MSCs from the original stocks (Passage 0) were cultured until they reached 90% confluency. The MSCs were then transfected with 12 nM *Hgf* Silencer Select predesigned siRNA (ID: s127876) or Silencer Select negative control No. 1 siRNA (Thermo Fisher Scientific, MA, USA) using Lipofectamine RNAiMAX transfection reagent (Thermo Fisher Scientific). At 24 h after the start of siRNA transfection, the cells were harvested and cryopreserved in CP-1 High Grade cryoprotectant until use.

### Analysis of Hgf expression

Total RNA was isolated from synovial MSCs transfected with either *Hgf* siRNA or negative control siRNA using the RNeasy Mini Kit (Qiagen, Venlo, Netherlands). The cells used for RNA extraction were obtained from a cryopreserved stock that had been frozen 24 h after siRNA transfection. Complementary DNA (cDNA) was synthesized using the High Capacity RNA-to-cDNA Kit (Thermo Fisher Scientific). Quantitative PCR was performed in triplicate for each sample using the TaqMan Gene Expression Master Mix (Thermo Fisher Scientific) and a CFX384 Touch Real-Time PCR Detection System (Bio-Rad Laboratories, Hercules, CA), according to the manufacturer's instructions.

TaqMan™ gene expression assays were used to quantify rat *Hgf* (Assay ID: Rn00566673_m1) and the endogenous control *Actb* (Assay ID: Rn00667869_m1) (Thermo Fisher Scientific). Gene expression levels were calculated using the 2^–ΔΔCt method, with normalization to *Actb*. The synovial MSCs used in this study were pooled from 15 individual ACI rat donors, and all gene expression analyses were conducted using this single pooled cell population, representing one biological replicate. Quantitative PCR was performed in technical triplicates, and the values presented in the figures reflect these technical replicates.

### Analysis of HGF protein secretion

HGF protein secretion was assessed using a Rat HGF ELISA Kit (Abnova, Taipei, Taiwan) in synovial MSCs transfected with either *Hgf* siRNA or negative control siRNA. The cryopreserved cells were thawed, seeded, and cultured in cell isolation medium for up to 7 days. Culture supernatants were collected on days 1, 3, and 7. Prior to analysis, the supernatants were centrifuged at 300 × *g* for 5 min to remove cell debris and stored at –80°C. Undiluted supernatants were used for ELISA, and culture medium without MSCs served as a negative control.

The ELISA procedure was performed according to the manufacturer's protocol. Briefly, 100 µL of standards or samples was added to each microplate well and incubated for 2 h at room temperature. A 100 µL volume of biotinylated detection antibody was then added to each well, followed by a 2 h incubation at room temperature (RT). After washing, 100 µL of streptavidin–peroxidase conjugate was added and incubated for 40 min at room temperature. Following another wash, 90 µL of chromogen substrate was added. After a 7 min incubation, the reaction was stopped with 100 µL of stop solution. Absorbance was measured at 450 nm using a Cytation 5 plate reader (BioTek, Agilent Technologies, Santa Clara, CA).

As with the gene expression analysis, ELISA assays were performed in technical triplicates using supernatants derived from the same pooled MSC batch (one biological replicate).

## In vitro chondrogenesis assay

Synovial MSCs transfected with either *Hgf* siRNA or negative control siRNA were used to assess chondrogenic differentiation. Cells were derived from a single cryovial of pooled, cryopreserved MSCs, representing one biological replicate. For each group, three chondrogenic pellets were prepared as technical replicates (n = 3 per group) by suspending $2.5 \times 10^5$ cells per pellet in 1 mL of chondrogenic medium. The chondrogenic medium consisted of high-glucose DMEM (Thermo Fisher Scientific) supplemented with 10 ng/mL TGF-β3 (R&D Systems Inc.), 3.92 µg/mL dexamethasone (FUJIFILM Wako Pure Chemical Corp., Osaka, Japan), 50 µg/mL L-ascorbic acid 2-phosphate (Cayman Chemical Company, Ann Arbor, MI, USA), 40 µg/mL L-proline (MP Biomedicals, Irvine, CA, USA), 1 µg/mL sodium pyruvate (Thermo Fisher Scientific), 1% ITS-X supplement (100×; FUJIFILM Wako Pure Chemical Corp.), and 0.5 µg/mL BMP-2 (R&D Systems Inc.). The cell suspension was transferred to 15 mL polypropylene tubes and centrifuged at $450 \times g$ for 10 min to form pellets. Pellets were then cultured for 3 weeks, with medium changes every 3–4 days [11,12].

For histological evaluation, the pellets were fixed, embedded in paraffin, sectioned at 5 µm thickness, and stained with safranin O and toluidine blue. For immunohistochemistry, type II collagen was detected using a purified anti-human type II collagen IgG antibody (clone II-4C11; Kyowa Pharma Chemical Co., Ltd., Toyama, Japan) diluted 1:200 in Antibody Diluent (DAKO, Glostrup, Denmark). The secondary antibody was Histofine Simple Stain Mouse MAX-PO(R) (Nichirei Corp., Tokyo, Japan). Sections were incubated with the primary antibody for 60 min at room temperature, followed by incubation with the secondary antibody for 30 min at room temperature. Immunoreactivity was visualized using 3,3′-diaminobenzidine (DAB; Muto Pure Chemicals Co., Ltd., Tokyo, Japan) as the chromogen.

## Anterior cruciate ligament transection (ACLT) surgery and MSC Injection

Following an acclimation period of at least 7 days, bilateral anterior cruciate ligament (ACL) transection was performed in 30 Lewis rats by lateral dislocation of the patella to expose the joint, as previously described [3]. All surgeries were conducted under isoflurane anesthesia. Anesthesia was induced with 3–5% isoflurane in an induction chamber and maintained at 1–3% via a nose cone. Body temperature was maintained throughout the procedure, and the anesthesia depth and the general condition of the animals were continuously monitored. After surgery, the rats were allowed free movement in their cages. To minimize procedural pain, surgeries were performed under adequate anesthesia by an experienced operator. Postoperative monitoring was carried out daily, and no animals showed signs of severe pain or distress.

One week after surgery, the rats were randomly divided into three groups of 10 animals each (n = 10), resulting in 20 knee joints per group (n = 20). Under isoflurane anesthesia, a 28-gauge needle was used for intra-articular injection of 50 µL of the following assigned treatments into the knee joints of the animals in each group:

1. *Hgf* siRNA-transfected MSCs ($1 \times 10^6$ cells),

2. Negative control siRNA-transfected MSCs ($1 \times 10^6$ cells), or

3. Vehicle alone (CP-1 High Grade cryoprotectant without MSCs).

Each treatment was administered once weekly for 7 consecutive weeks. MSCs were thawed immediately before injection, and cell viability was confirmed to be ≥ 90% at the time of administration.

The sample size was based on prior in vivo studies [3] to ensure sufficient power for detecting group differences. Of the 34 animals initially purchased, 4 were excluded prior to surgery due to health concerns and were euthanized in accordance with institutional guidelines.

## Macroscopic evaluation

At 8 weeks after ACLT surgery, all animals were euthanized by exsanguination through abdominal aortic resection under 2.5% isoflurane inhalation anesthesia administered via mask. The tibial and femoral condyles were harvested separately and stained with India ink for macroscopic observation. Cartilage degeneration was assessed by an observer blinded to the treatments using a macroscopic scoring system ranging from 0 to 6, based on the severity of damage, as described by Horiuchi et al. [13]. The scoring criteria were as follows:

# Grade 0 Intact articular surface

# Grade 1 Fibrillation (<0.5 mm)

# Grade 2 Fibrillation (≥0.5 mm)

# Grade 3 Width of erosion area (<0.5 mm)

# Grade 4 Width of erosion area (≥0.5 mm, <1 mm)

# Grade 5 Width of erosion area (≥1 mm, <1.5 mm)

# Grade 6 Width of erosion area (≥1.5 mm, <2 mm)

For each rat, the medial tibial plateau and femoral condyle were scored separately, with the higher score from these two regions used in the final analysis.

## Histological examination

Histological examinations were performed by fixing both tibial and femoral cartilage samples in a 10% formalin neutral buffer solution for 2 days, followed by decalcification with 20% ethylenediaminetetraacetic acid (EDTA; Fujifilm Wako, Tokyo, Japan) for two weeks. The samples were then embedded in paraffin wax. The medial condyle specimens were sectioned sagittally at 5 μm thickness and stained with Safranin O and Fast Green for analysis. Cartilage degeneration was assessed by an observer blinded to the treatments using the Osteoarthritis Research Society International (OARSI) scoring system, which provides a detailed grading methodology [14]. This system includes six main grades (0–6) with corresponding subgrades:

# Grade 0: Intact surface and cartilage with no subgrade distinctions.

# Grade 1: Intact surface, with subgrade 1.0 indicating intact cells and 1.5 indicating cell death.

# Grade 2: Surface discontinuity, with subgrade 2.0 showing fibrillation through the superficial zone and 2.5 indicating surface abrasion with matrix loss in the superficial zone.

# Grade 3: Vertical fissures, with subgrade 3.0 showing simple fissures and 3.5 indicating branched or complex fissures.

# Grade 4: Erosion, with subgrade 4.0 showing superficial zone delamination and 4.5 indicating mid-zone excavation.

# Grade 5: Denudation, with subgrade 5.0 showing intact bone surface and 5.5 indicating the presence of reparative tissue.

# Grade 6: Deformation, with subgrade 6.0 showing joint margin osteophytes and 6.5 indicating both joint margin and central osteophytes.

The cartilage surface involvement was also evaluated using the OA cartilage histopathology—stage assessment, which defines the extent of joint involvement (% surface area) as follows:

# Stage 0: No OA activity seen.

# Stage 1: <10%.

# Stage 2: 10–25%.

# Stage 3: 25–50%.

# Stage 4: >50%.

The histological score was calculated as the product of the grade (depth progression into cartilage) and the stage (extent of joint involvement), resulting in a comprehensive score ranging from 0 to 26 points (where 0 indicates no joint involvement and 26 represents severe joint involvement).

## Statistical analysis

Statistical analyses were performed using GraphPad Prism version 5.04 (GraphPad Software Inc., San Diego, CA, USA). Nonparametric methods were used throughout this study to avoid assumptions about data distribution and to ensure robustness, particularly given the small sample sizes.

Group differences were first assessed using the Kruskal–Wallis test. When significant results were obtained, pairwise comparisons were performed using Mann–Whitney U tests with Bonferroni correction for multiple comparisons. The Bonferroni method was selected to control the family-wise error rate. Accordingly, the significance threshold was adjusted to $p < 0.0167$ (0.05 divided by 3 comparisons).

Statistical significance in the results was indicated as follows: $p < 0.0167$, $p < 0.0033$, $p < 0.00033$, depending on the number of comparisons made. All data are presented as medians with interquartile ranges. A p-value $< 0.0167$ was considered statistically significant.

## Results

### *Hgf* knockdown

Synovial MSCs treated with *Hgf* siRNA showed reduced *Hgf* gene expression, with levels only 0.304-fold those of cells treated with control siRNA (Fig 1a). While synovial MSCs treated with control siRNA showed an increase in HGF protein levels on days 3 and 7, MSCs treated with *Hgf* siRNA showed levels similar to the medium-only controls (negative control), indicating suppression of HGF secretion through day 7 (Fig 1b).

### In vitro analysis

Macroscopic observation of cartilage pellets after 3 weeks of differentiation (n = 3 per group) revealed that *Hgf* siRNA-treated pellets tended to be smaller in both weight and diameter compared to control siRNA-treated pellets (Fig 2). Statistical analysis was not performed due to the small sample size and the fact that pellets were derived from a single pooled MSC batch (biological replicate = 1). These properties were considered insufficient to yield reliable statistical conclusions.

Histological analysis showed that both groups produced cartilage matrix that stained positively with toluidine blue, safranin O, and type II collagen (detected by immunohistochemistry using DAB as the chromogen). However, the staining intensity of safranin O appeared weaker in the *Hgf* siRNA-treated group, and the area of matrix staining relative to the overall pellet was slightly reduced. These findings suggest that HGF secreted by synovial MSCs may influence the matrix production capacity during chondrogenic differentiation.

### Knee findings after cell administration

The induction of immune responses following the administration of MHC-mismatched cells was assessed by examining knee swelling and joint tissue hypertrophy during postmortem examination after synovial MSC administration. However, joint swelling and tissue thickening were observed uniformly across all groups and were not specific to the cell-administered group.

a

| | Mean Ct value (in triplicate) | | ΔCt value (Ct$_{Hgf}$-Ct$_{Actb}$) | ΔΔCt value (ΔCt HGF si - ΔCt Cont si) | Fold change (2^-ΔΔCt) |
|---|---|---|---|---|---|
| | Hgf | Actb | | | |
| Cont si | 36.558 | 21.975 | 14.583 | - | 1 |
| HGF si | 38.156 | 21.872 | 16.303 | 1.720 | 0.304 |

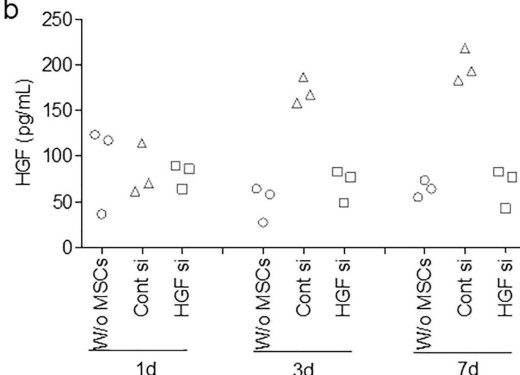

**Fig 1. Suppression of HGF secretion in synovial MSCs using *Hgf* siRNA.** (a) Quantitative PCR analysis of *Hgf* gene expression. Values represent technical triplicates from a single pooled MSC sample (biological replicate = 1). The relative expression level of the control siRNA-treated group (Cont si) was set to 1. (b) Quantification of HGF protein secretion by synovial MSCs using ELISA. W/o MSCs indicates culture medium without seeded cells (negative control). Values represent technical triplicates from the same pooled MSC batch..

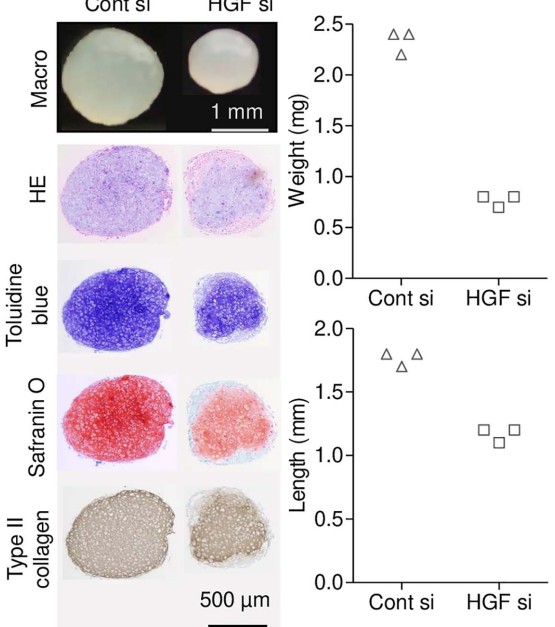

**Fig 2. Effect of HGF suppression on chondrogenic differentiation of synovial MSCs in vitro.** Macroscopic and histological images showing cartilage pellet morphology after chondrogenic differentiation of synovial MSCs treated with control siRNA (Cont si) or Hgf siRNA (HGF si). Histological staining included hematoxylin and eosin (HE), Toluidine blue, and Safranin O. Type II collagen was detected by immunohistochemistry using DAB as the chromogen. Graphs indicate the quantification of pellet weight (mg) and length (mm). Each point represents a single pellet (n = 3).

## Macroscopic analysis

Macroscopic observation at 8 weeks after ACL transection (Fig 3) revealed that the femoral cartilage with vehicle injection (without MSCs) showed fibrillation in 1 knee and erosion in 19 knees (Fig 4). The group receiving weekly injections of Control siRNA-treated MSCs for 7 weeks showed fibrillation in 5 knees and erosion in 15 knees. The *Hgf* siRNA-treated MSC group showed fibrillation in 4 knees and erosion in 16 knees.

Tibial cartilage injected with vehicle showed fibrillation in 1 knee and erosion in 19 knees (Fig 5). The Control siRNA-treated MSC group showed fibrillation in 4 knees and erosion in 16 knees, whereas the *Hgf* siRNA-treated MSC group showed no fibrillation in any knees but erosion in all 20 knees.

The lesions were scored on a scale of 0–6, with fibrillation classified into 2 grades and erosion into 4 grades based on their lengths. In the femoral cartilage, the median macroscopic score was lower in the Control siRNA group than in the vehicle group, although the difference was not statistically significant (Fig 6). Between the Control siRNA and *Hgf* siRNA groups, only 2 knees in the *Hgf* siRNA group showed a score of 1 point, compared to 5 knees in the Control siRNA group. In the tibial cartilage, the macroscopic score was significantly lower in the Control siRNA group than in the vehicle

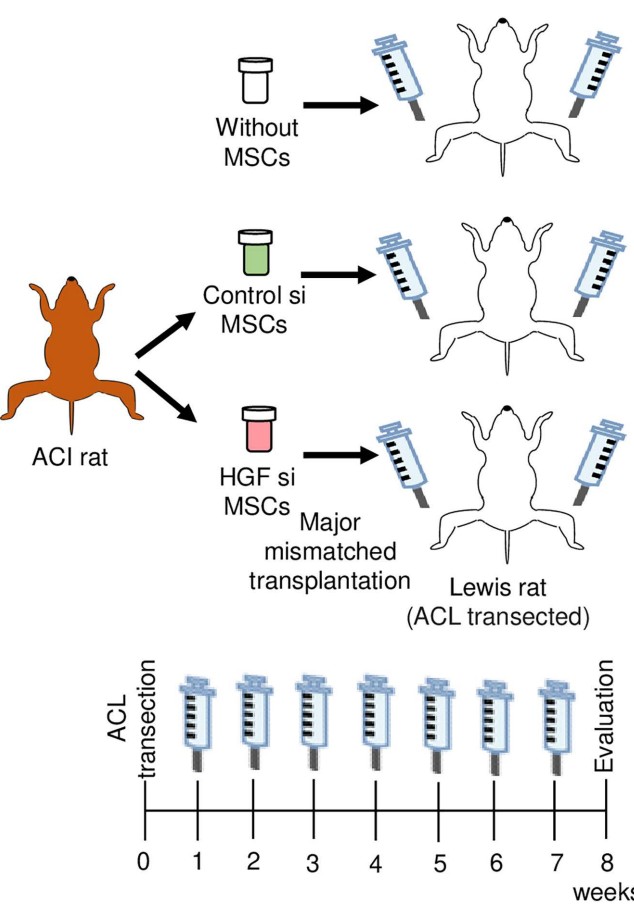

**Fig 3. Experimental protocol for major mismatched transplantation of synovial MSCs in a rat ACL injury model.** Synovial MSCs ($1 \times 10^6$ cells) treated with either negative control siRNA (Cont si) or *Hgf* siRNA (HGF si) were injected into the knee. Cryopreservation medium without MSCs served as a vehicle control. Weekly intra-articular injections were performed for 7 weeks, starting at 1 week after ACL transection, followed by evaluation at week 8.

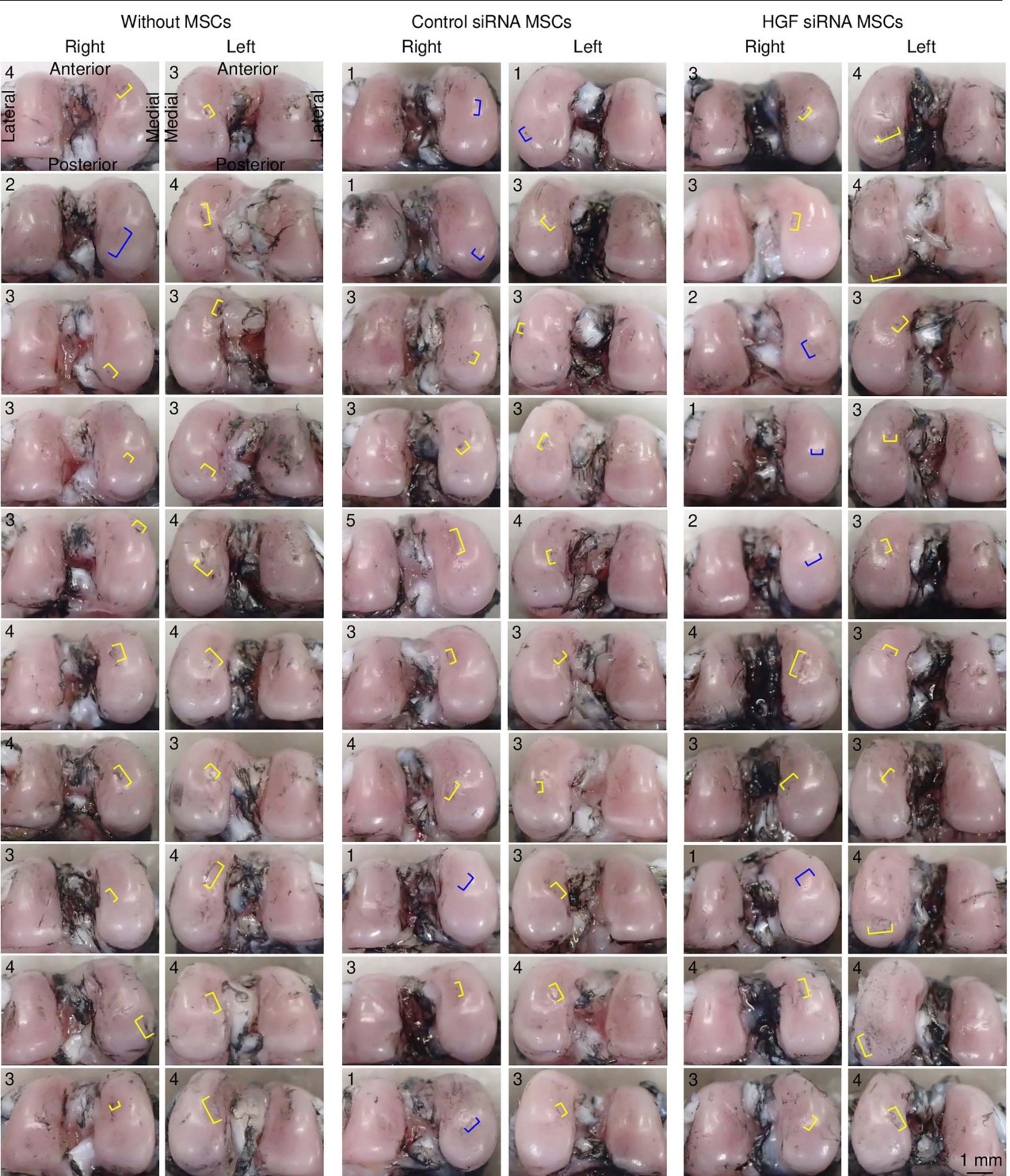

**Fig 4. Femoral cartilage lesions with macroscopic evaluation of all specimens.** India ink staining showing the articular cartilage surface lesions of femoral condyles in the three groups. Areas of fibrillation are indicated with blue brackets, and areas of erosion are indicated with yellow brackets. Numbers in each image represent macroscopic scores.

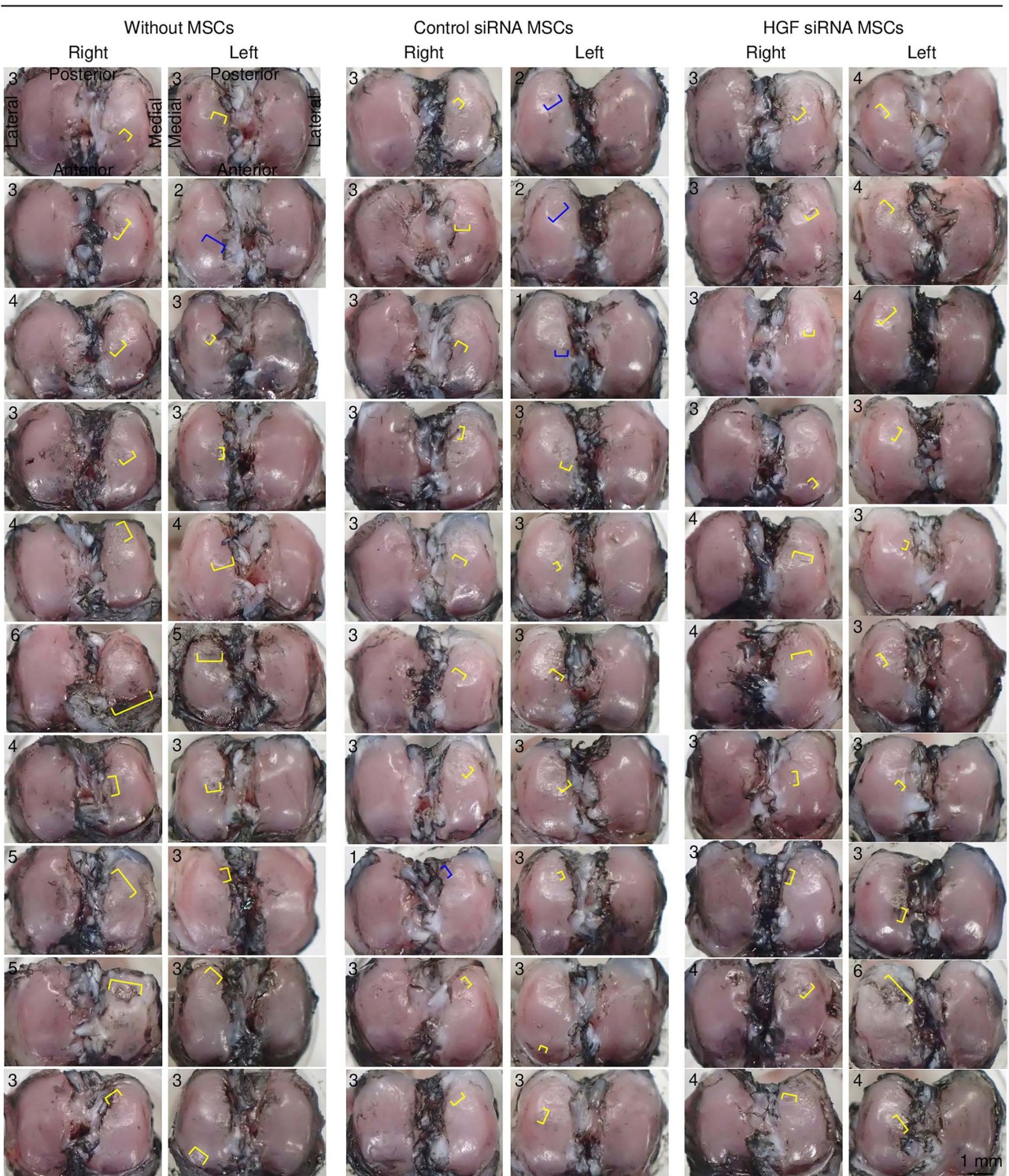

**Fig 5. Tibial cartilage lesions with macroscopic evaluation of all specimens.** India ink staining demonstrates the articular cartilage surface lesions of femoral condyles in three groups. Areas of fibrillation are indicated with blue brackets, and areas of erosion are indicated with yellow brackets. Numbers in each image represent macroscopic scores.

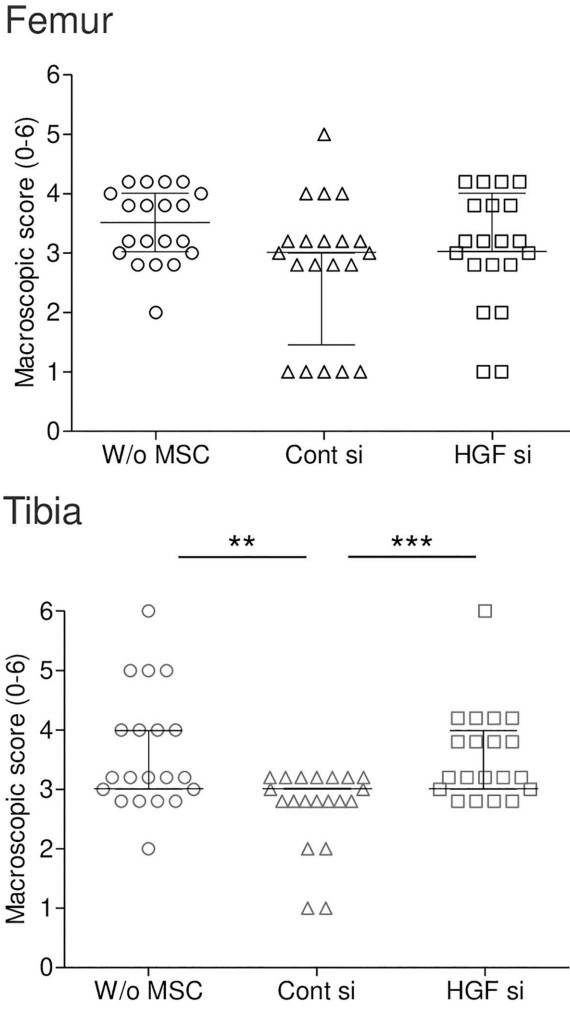

**Fig 6. Quantitative analysis of macroscopic scores (0–6) for cartilage lesions.** Midlines and error bars representing the median ± interquartile range (IQR) (n = 20). Midlines represent median values and error bars represent interquartile range (IQR) (n = 20). Femur scores: W/o MSC: 3.5 (IQR: 3.0–4.0); Cont si: 3.0 (IQR: 1.5–3.0); HGF si: 3.0 (IQR: 3.0–4.0). Tibia scores: W/o MSC: 3.0 (IQR: 3.0–4.0); Cont si: 3.0 (IQR: 3.0–3.0); HGF si: 3.0 (IQR: 3.0–4.0). **p < 0.0033, ***p < 0.00033 between two groups.

group (p < 0.0033), whereas the score was significantly higher in the *Hgf* siRNA group than in the Control siRNA group (p < 0.00033).

## Histological analysis

For the femoral cartilage, each specimen showed characteristic Safranin O-stained cartilage layers with varying degrees of damage (Fig 7). The OARSI histological scores ranged from 2 to 10.5 across all specimens. The overall pattern of cartilage damage appeared relatively consistent across all treatment groups.

For the tibial cartilage, notable differences were observed in the OARSI scores, which ranged from 3.5 to 26. The Without MSCs group showed consistently higher scores (Fig 8). The lesion patterns in the tibial plateaus demonstrated greater variability compared to the femoral condyles, with some specimens exhibiting extensive damage in agreement with their higher OARSI scores.

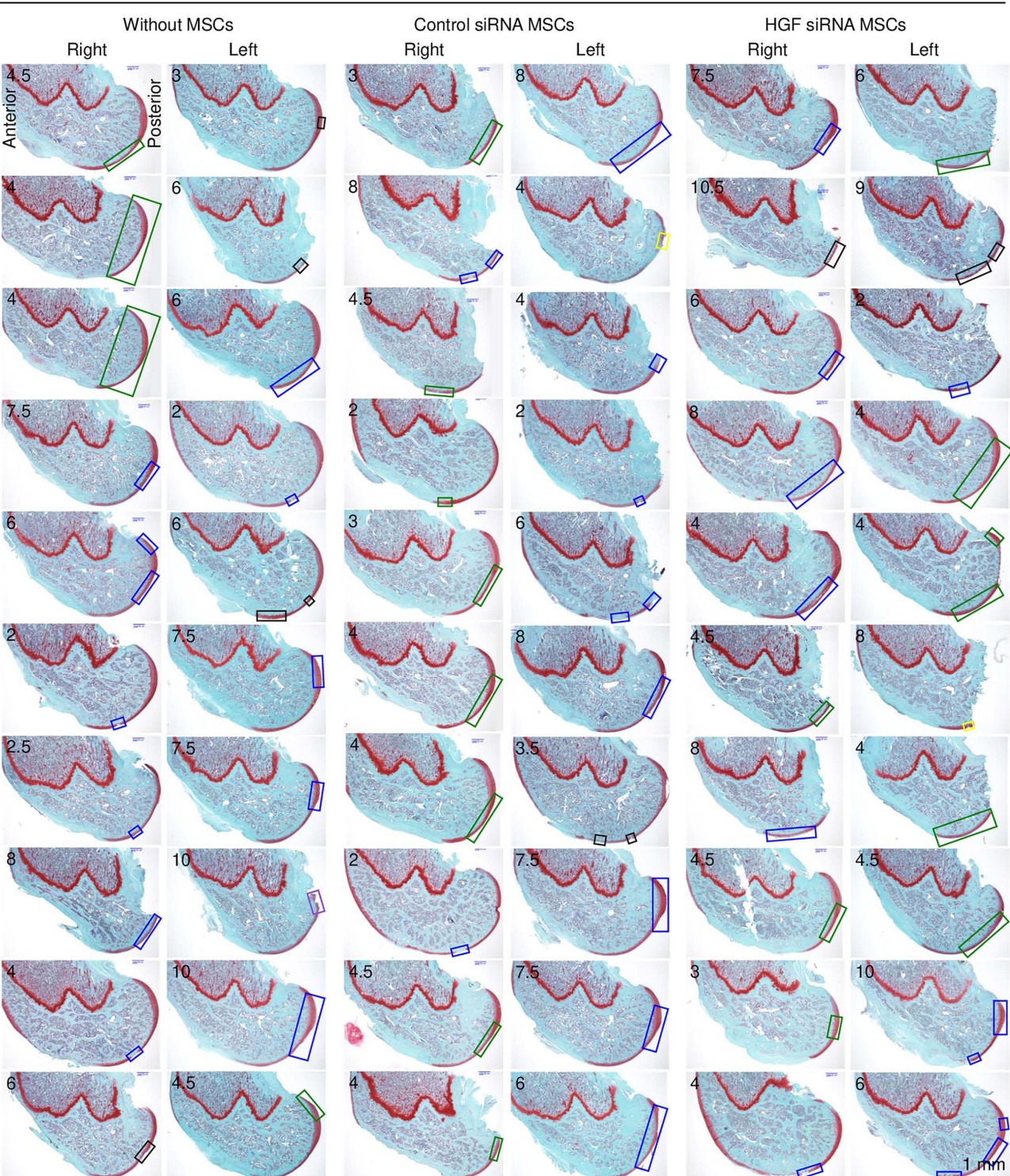

**Fig 7. Femoral cartilage lesions with microscopic evaluation of all specimens.** Medial femoral condyles were sectioned in the sagittal plane and stained with Safranin O. The length of the square corresponds to the lesion length, and its color indicates the OARSI histopathology subgrade (depth damage): green for grades 1 or 1.5, blue for 2 or 2.5, black for 3 or 3.5, yellow for 4 or 4.5, purple for 5 or 5.5, and red for 6 or 6.5. The OARSI histological scores are shown in each image.

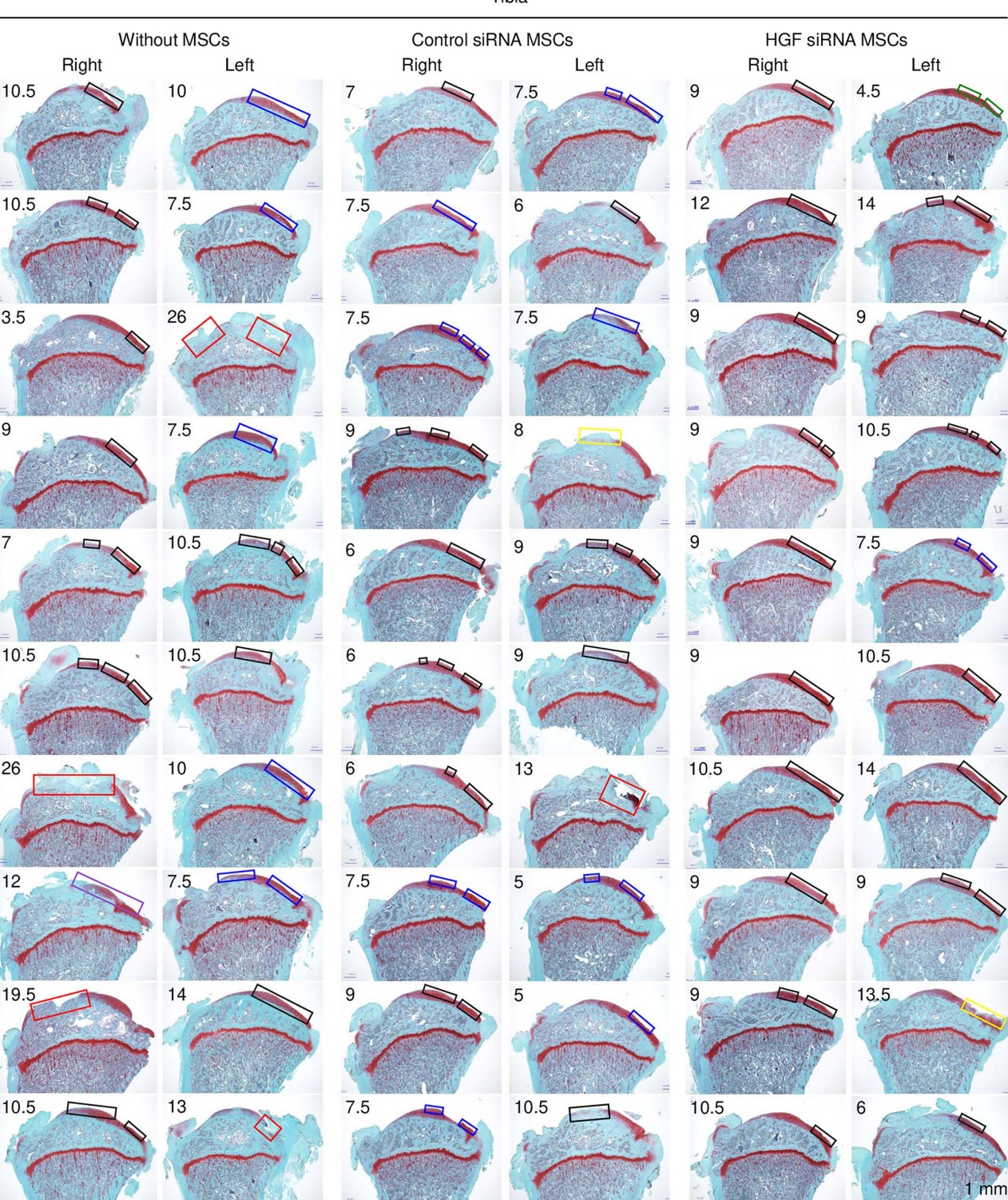

**Fig 8. Tibial cartilage lesions with microscopic evaluation of all specimens.** Medial tibial plateaus sectioned in the sagittal plane and stained with Safranin O. The length of the square corresponds to the lesion length, and its color indicates the OARSI histopathology subgrade (depth damage): green for grades 1 or 1.5, blue for 2 or 2.5, black for 3 or 3.5, yellow for 4 or 4.5, purple for 5 or 5.5, and red for 6 or 6.5. The OARSI histological scores are shown in each image.

Quantitative analysis revealed that the median scores and interquartile ranges in the femoral cartilage were similar across all three groups, with scores clustering around 4–6 and showing no statistically significant differences (Fig 9). In contrast, the tibial cartilage analysis demonstrated significant differences between the groups. The scores were significantly lower for the Control siRNA MSCs group than for the Without MSCs group ($p < 0.0033$). Conversely, the scores were significantly higher for the *Hgf* siRNA MSCs group than for the Control siRNA MSCs group ($p < 0.0033$).

## Discussion

This study evaluated the role of hepatocyte growth factor (HGF) in the chondroprotective effects of synovial mesenchymal stem cells (MSCs) under fully MHC-mismatched allogeneic conditions. Using a rigorous ACI-to-Lewis rat OA model, we demonstrated that synovial MSCs retained their therapeutic efficacy despite the complete mismatch in MHC class I and II antigens. Importantly, siRNA-mediated knockdown of Hgf impaired both in vitro chondrogenic differentiation and in vivo cartilage protection. These findings suggest that the beneficial effects of synovial MSCs in this immunologically challenging setting are at least partially dependent on HGF, which appears to support cartilage matrix production and maintain MSC functionality.

## Femur

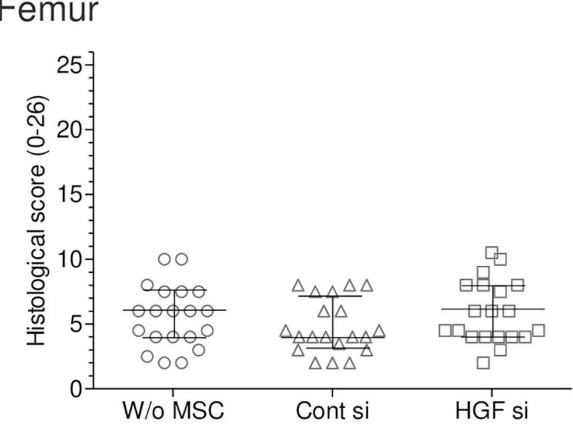

## Tibia

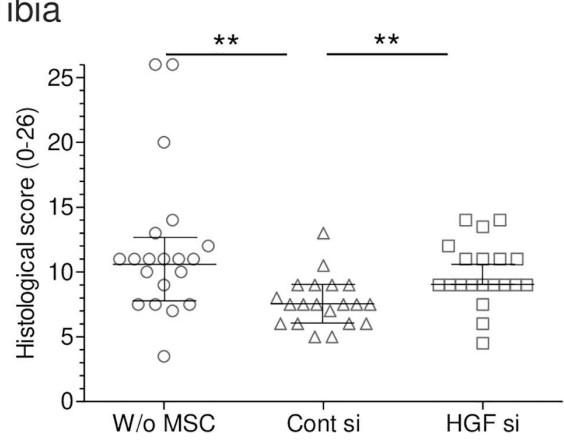

**Fig 9. Quantitative analysis of OARSI histological scores (0-26).** Midlines represent median values and error bars represent interquartile range (IQR) (n = 20). Femur scores: W/o MSC: 6.0 (IQR: 4.0-7.5); Cont si: 4.0 (IQR: 3.1-7.1); HGF si: 5.2 (IQR: 4.0-8.0). Tibia scores: W/o MSC: 10.5 (IQR: 7.8-12.7); Cont si: 7.5 (IQR: 6.0-9.0); HGF si: 9.0 (IQR: 9.0-10.5). **$p < 0.0033$ between two groups.

Intra-articular injection of synovial MSCs demonstrated chondroprotective effects in tibial cartilage but not in femoral cartilage in the rat ACL transection model. This result likely reflected the more severe and earlier degeneration observed in the tibial cartilage in this model. Following ACL transection, meniscal extrusion occurs, leading to a concentration of excessive mechanical stress on the articular cartilage, particularly in the region adjacent to the inner margin of the meniscus. Due to anatomical features, this stress remains more localized on the tibial side than on the femoral side during repeated knee flexion–extension movements. Consequently, cartilage degeneration is most likely to initiate in the tibial plateau. According to the 3D MRI analysis of human knees by Katano et al., medial meniscus extrusion in medial osteoarthritis triggers cartilage wear along the inner margin of the medial meniscus in the tibial cartilage [15]. Furthermore, femoral cartilage degeneration occurs either as wear along the inner margin of the medial meniscus or independently as a mirror lesion of the tibial cartilage wear due to medial meniscus extrusion [16]. Thus, the early tibial cartilage degeneration observed in the rat ACL transection model is likely to be the primary reason why synovial MSC injection effects are limited to the tibial cartilage. This phenomenon can be attributed to the concentration of anatomical and biomechanical stress in the knee joint. Nonetheless, the effects observed on the tibial cartilage were clearly dependent on the ability of the MSCs to secrete HGF.

Among the growth factors produced by synovial MSCs, we focused on HGF in this study because, as a representative cytokine expressed by MSCs, HGF has a known involvement in cell proliferation and migration [17]. HGF promotes cartilage matrix production [8] and is suggested to be an essential factor for chondrogenic differentiation. Wakitani et al. demonstrated that direct injection of HGF molecules into the knees of cartilage-defect model animals restored defective cartilage [17]. Thus, HGF may play a crucial role in cartilage regeneration through its multiple functions, and HGF production and secretion may function as key mechanisms underlying the chondroprotective effects of synovial MSCs.

The Hgf-knockdown MSCs produced smaller pellets than were obtained using control MSCs. In MSC pellet culture, the enlargement of cell aggregates is attributed to cartilage matrix production rather than to an increase in cell number [18]; hence, the size of the cartilage pellets reflects the capacity of the MSC populations to produce cartilage matrix [12]. Therefore, the reduction in pellet size observed with Hgf knockdown indicates that HGF plays a crucial role in cartilage matrix production by synovial MSCs. Hgf knockdown attenuated the therapeutic effects of MSCs in the in vivo OA model rats, highlighting the importance of this growth factor in MSC-mediated cartilage production.

The chondroprotective effects of synovial MSC injections in OA are mediated through complex paracrine mechanisms rather than via direct cellular differentiation. Following intra-articular injection, MSCs primarily localize to the synovium, where they maintain their stem cell properties [3] and orchestrate therapeutic actions by secreting multiple bioactive factors. These secreted mediators include HGF, TSG-6 (which exerts anti-inflammatory effects [19]), PRG-4 (lubricin; which enhances joint lubrication and cartilage homeostasis [20]), and BMP-2/6 (which promotes chondrocyte differentiation and matrix synthesis [21]). Together, these factors create a comprehensive therapeutic milieu that simultaneously addresses the key pathological features of OA: inflammation, lubrication deficiency, and cartilage degeneration [22].

In the present study, Hgf knockdown in synovial MSCs resulted in a moderate but statistically significant reduction in both in vitro chondrogenic capacity and in vivo therapeutic efficacy, demonstrating that HGF contributes meaningfully to the overall chondroprotective effects of MSCs. However, since other key paracrine mediators were not evaluated in this study, the relative contribution of HGF within the broader multimodal therapeutic framework remains to be fully elucidated. Future studies examining the individual and synergistic roles of various secreted factors will be necessary to obtain a comprehensive understanding of the mechanistic hierarchy underlying MSC-mediated cartilage protection.

Early passage synovial MSCs (passages 0 or 1) were selected for both in vitro and in vivo experiments. This decision was based on previous studies by Okuno et al. [6] and Ozeki et al. [3], which also employed early passage cells. Clinical applications of synovial MSCs typically involve cells at passage 0 or 1 [4,23–25], since extended passaging can reduce MSC stemness and regenerative capacity [12].

Several limitations of this study should be noted. One was that we used the anterior cruciate ligament transection (ACLT) model, which, although widely adopted for post-traumatic osteoarthritis research, may not fully recapitulate the complex pathology of human OA. While this model allows for reproducibility and controlled OA induction, its translational relevance remains limited [26].

Another limitation was that we administered synovial MSCs via intra-articular injection once weekly for 7 weeks, which differs from typical clinical protocols where such frequent dosing may not be feasible [4]. Future studies should explore more clinically relevant dosing regimens.

We also did not track the persistence or migration of injected cells, which is important for evaluating therapeutic durability and safety [3]. Additionally, detailed safety assessments of allogeneic transplantation, including long-term immune responses [27] and potential ectopic tissue formation [28], were not conducted and warrant future investigation.

The present analysis did not include sex-based cell tracking, although male donor rats and female recipient rats have been used in the past to enable the potential tracking of donor cells using Y chromosome-specific markers [29]; However, our aim was not to evaluate sex-specific effects, and this limitation should be considered when interpreting the results.

Our mechanistic analysis focused specifically on HGF function through siRNA knockdown. While this approach demonstrated the contribution of HGF to MSC efficacy, we did not conduct a comprehensive evaluation of whether *Hgf* knockdown affected other chondroprotective factors, such as PRG4 or BMP2/6 [3], nor did we perform rescue experiments to isolate the specific contribution made by HGF. Future studies incorporating secretome analysis and rescue experiments would provide more definitive mechanistic insights.

Finally, our in vivo outcome assessment relied primarily on histological evaluation. Including biochemical markers of cartilage composition and comprehensive analysis of synovial tissue responses could provide additional mechanistic insights into HGF-mediated chondroprotection.

## Conclusions

We have established that synovial MSCs can effectively function as therapeutic agents for OA treatment across major histocompatibility barriers, with HGF identified as an essential molecular mediator of their chondroprotective effects. The successful therapeutic outcomes observed in the fully MHC-mismatched ACI-to-Lewis rat model demonstrate the clinical potential of allogeneic MSC therapy, especially for expanding donor availability and treatment accessibility. Our mechanistic analysis through HGF knockdown definitively confirmed that HGF secretion is indispensable for MSC-mediated cartilage protection, thereby establishing this growth factor as a key therapeutic target. Our results support the clinical translation of allogeneic synovial MSC therapy as an OA treatment and suggest that HGF pathway modulation could enhance therapeutic outcomes in regenerative medicine approaches for cartilage disorders.

## Supporting information

**S1 File. The ARRIVE guidelines 2.0: author checklist.**
(DOCX)

## Acknowledgments

We thank Ms. Ellen Roider for English editing. The authors declare that they have not used AI-generated work in this manuscript.

## Author contributions

**Conceptualization:** Tsukasa Kitahashi, Kentaro Nakamura, Ichiro Sekiya.
**Data curation:** Tsukasa Kitahashi, Ichiro Sekiya.

**Formal analysis:** Tsukasa Kitahashi, Ichiro Sekiya.

**Funding acquisition:** Kentaro Nakamura.

**Investigation:** Tsukasa Kitahashi, Ryo Kogawa.

**Methodology:** Tsukasa Kitahashi.

**Project administration:** Tsukasa Kitahashi, Kentaro Nakamura.

**Resources:** Tsukasa Kitahashi, Kentaro Nakamura, Ryo Kogawa.

**Supervision:** Kentaro Nakamura, Ichiro Sekiya.

**Validation:** Kentaro Nakamura, Ryo Kogawa.

**Visualization:** Tsukasa Kitahashi, Ichiro Sekiya.

**Writing – original draft:** Tsukasa Kitahashi.

**Writing – review & editing:** Kentaro Nakamura, Ichiro Sekiya.

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
