## [Decision Letter · Decision Letter 0]

3 Jul 2025

Dear Dr. Sekiya,

Thank you for submitting your manuscript to PLOS ONE. After careful consideration, we feel that it has merit but does not fully meet PLOS ONE’s publication criteria as it currently stands. Therefore, we invite you to submit a revised version of the manuscript that addresses the points raised during the review process.

We look forward to receiving your revised manuscript.

Kind regards,

Xindie Zhou

Academic Editor

PLOS ONE

Journal Requirements:

“Tsukasa Kitahashi, Kentaro Nakamura, and Ryo Kogawa are employees of FUJIFILM Corporation. The remaining authors declare no competing interests.”

We note that one or more of the authors are employed by a commercial company: FUJIFILM Corporation

4. We note that your Data Availability Statement is currently as follows: All relevant data are within the manuscript and in Supporting Information files.

Reviewers' comments:

Reviewer's Responses to Questions

**Comments to the Author**

1. Is the manuscript technically sound, and do the data support the conclusions?

Reviewer #1: Yes

Reviewer #2: Yes

2. Has the statistical analysis been performed appropriately and rigorously?

Reviewer #1: Yes

Reviewer #2: No

3. Have the authors made all data underlying the findings in their manuscript fully available?

Reviewer #1: Yes

Reviewer #2: Yes

4. Is the manuscript presented in an intelligible fashion and written in standard English?

Reviewer #1: Yes

Reviewer #2: Yes

Reviewer #1: The study asks whether major histocompatibility mismatch in synovial-derived MSCs will block the chondroprotective effect previously shown by the same group in a different rat OA induction model.

The study also asks whether the synovial-derived MSC’s ability to express HGF is necessary for their therapeutic effect.

The major findings of the study are that synovial derived MSCs appear to still have a chondroprotective effect despite MHC mismatch; and that this effect was not observed if synovial derived MSCs with siRNA induced HGF knockdown were used.

Overall comments:

The strengths of the study are the well-controlled designs, reproducibility obtained, and careful analyses. The weaknesses of the study are the lack of translational relevance of the ACLT model as a clinical model of human OA induction in which damage is much more common in the femoral condyle; the lack of translational relevance of the MSC injection regimen as a clinical injection regimen where repeated injections would not be feasible; and lack of consideration of other signals affected by HGF knockdown in the MSCs that may be impairing their activity versus the author’s inference that the chondroprotective activity of the cells is due exclusively to HGF knockdown.

Specific comments:

Describe size of needle used in MSC injections.

Describe if the MHC mismatched cell injections led to any immune response in the animal – eg, presence of lymphocytes in the synovium should be examined

The authors have previously identified Prg4 and BMP2,6 as additional potentially pro-chondrogenic factors expressed by synovial MSCs. Were the levels of Prg4 and BMP2,6 mRNA and protein also decreased in the cells by HGF knockdown? Without this information, it is an overstatement to attribute the chondroprotective

Reviewer #2: This manuscript presents a study aimed at determining the implication of hepatocyte growth factor (HGF) in the therapeutic effects of synovial derived mesenchymal stem cells (MSCs) in a rat osteoarthritis (OA) model. To do this, both in vitro and in vivo studies have been conducted. First, HGF expression is silenced, and it is demonstrated that this silencing has a negative effect for the chondrogenic differentiation of synovial MSCs in vitro. Second, both HGF-silenced and control MSCs are administered into rat OA knees (as well as no-cells control), and the effects of these administrations is measured by macroscopic and histological findings in cartilage. Of note, the in vivo study is designed for the MSC administration to be MHC-mismatched, using two different rat strains as donors (ACI) and recipients (Lewis).

Overall, the manuscript is well written, the goals are clear (but could be more precisely defined), the study design and methods are appropriate (they could be complemented, though), and the results are generally clearly presented. Limitations are discussed and conclusions are based on the results. In general, this is an interesting and formally adequate manuscript, however some points need attention prior to further considering it for publication.

Please see below main general comments, and later more specific comments pointed out along the manuscript. I hope these points can help authors making the most of their manuscript, but if they disagree with any comment/suggestion, please kindly explain why.

General comments:

- I strongly recommend putting more emphasis into the novelty of the study. The introduction is very brief and, in my opinion, it does not provide enough context to understand the relevance of the present work. Previous studies have already assessed the role of HGF in MSC mechanisms in vitro and in vivo, and some of them have also assessed knockdown cells in different disease models, so the authors should answer very clearly in the introduction: what is the novelty of our study? What does our study adds to the current knowledge?

- To assess the effect of MHC-mismatched administration is not an actual goal of the study, but part of its design. While I fully agree that allogeneic MHC-mismatched administration is the most realistic scenario for clinical use, and thus it is important to assess MSCs in this scenario, this study does not really provide information about the effect of MHC-mismatching because no MHC-matched (or syngeneic) control is used. Thus, I recommend placing the focus towards accounting for this factor by selecting a mismatched scenario, but not towards unveiling the effect of such scenario. This should be reflected along the whole manuscript but especially in the goals (end of introduction), study design (beginning of the methodology) and conclusions.

- Evaluation of the in vivo study outcome mostly relies on histopathological assessment, and only cartilage is assessed. It could have helped elucidating HGF-mediated mechanisms to include more biochemical parameters about cartilage composition, and also assessing synovium as a key target of MSC therapy to modulate inflammation in OA. I consider that this should at least be discussed as a limitation.

- Linked to the previous point, it should be emphasized that this is an observational study but not a mechanistic one. This study reveals that HGF plays some role on the outcome of MSC therapy in OA, but does not provide information about which is this role. Thus, any claim about mechanisms should be done very carefully. In addition, regarding the in vitro part of the study, only chondrogenesis is assessed, however (as authors state in the discussion), MSC therapeutic effects in OA are mainly mediated via paracrine signalling and not by direct differentiation, so in vitro assays could have also been designed more in that direction. I consider that this should at least be discussed as a limitation.

Specific comments:

Introduction:

- In addition to what has been commented under general comments, I would recommend including in the introduction a brief explanation about the rational for choosing these two strains of rats (ACI and Lewis) to establish a MHC-mismatched transplantation scenario.

- Goals at the end of the introduction section should be adjusted as suggested under general comments, especially regarding the MHC-mismatching not being a goal itself but a study condition. Focus should be placed on the role of HGF in MSC therapeutics from an observational perspective.

Material and methods:

Animals:

- Any particular reason to use all male donors and all female recipients? Using half males and half females as recipients could have allowed to also accounting for the sex effect.

Isolation of rat synovial MSCs

- Under the previous section “Animals”, in lines 86-87, it is stated that "To establish MHC-mismatched synovial MSCs, 30 male ACI rats (...) were purchased", however here it is mentioned that 15 rats were used to create a pool of synovial MSCs. What the other 15 ACI male rats were used for?

- If I properly understood, all in vitro and in vivo assays were conducted with passage 0 and passage 1 cells. Were these passages selected for any particular reason? Early passages such as 0 and 1 may present higher cell heterogeneity compared to later passages, which are more frequently used for therapeutic purposes to provide a more homogeneous cell population.

- Were synovial MSCs characterised by any means? i.e immunophenotype, tri-lineage differentiation?

Analysis of Hgf expression

- At what time-points was HGF gene expression and protein secretion assessed? For gene expression it is stated "in the cryopreserved stock". Does it mean in the cells frozen after 24h of transfection? For protein secretion (ELISA), time points are not stated but in Results (figure 1) one can see day 1, 3 and 7. Please ensure that experiment design is explained with enough detail under methodology to match with what is presented under results.

- Since the synovial MSCs used consisted of a pool from 15 donors, I guess only this one biological (pooled) replicate was used, and that what is represented in figures are technical replicates. For example, for ELISA results (fig 1.C) three replicates are represented, but for gene expression it is not clear. Please explain these methodological details about biological and technical replicates under methodology section.

- Line 137: analysis method for gene expression should be 2^-ΔΔCt instead of 2^-ΔCt.

- Please briefly explain how the HGF ELISA was performed. Even if it was done according to kit's manufacturer's instructions, a brief explanation should be provided including any procedure on the supernatants (centrifugation, filtration, freezing...), any dilution of the samples, points in the standard curve, absorbance length and equipment used for absorbance reading, as well as negative controls used (i.e. what is represented as "w/o MSCs" in figure 1.C). This information cannot be exclusively placed in the figure captions but needs to be in the main text so the reader can get the necessary details on it.

In vitro chondrogenesis assay

- It is my understanding, according to the results (figure 2) that both HGF-silenced MSCs and control MSCs were used in chondrogeneic assays, however this is not detailed in the methodology. Please state it clearly in the methodology, the reader needs to get clear and direct information from the text and not to guess based on figures. Please also state the number of biological and technical replicates, as mentioned for gene expression/protein secretion of HGF.

- Please briefly describe the immunostaining procedure, even if previously described in bibliography (primary antibody clone, secondary antibody if applies, antibody dilution, etc.).

Anterior cruciate ligament transection (ACLT) surgery and MSC injection

- In line 89 (Animals section), it is stated that 34 female Lewis rats were purchased. Based on the abstract, 20 knees were included per group. According to this section, both knees of each rat were used. One can infer that 30 rats were used, divided into three groups of 10 rats per groups, and using both knees of each rat, thus making 20 knees per group. Two comments about this:

o This information about group distribution should be clearly explained in this section with exact number of animals and knees, and not left to be inferred by the reader from different parts of the manuscript.

o If 34 rats were purchased but 30 were used for the assay, what the other 4 rats were used for? One may guess that extra animals were purchased in case animals in the study needed to be replaced, but this should not be inferred but crystal clear explained. The destination of the unused/discarded animals (guess euthanasia?) should also be disclosed.

- Please briefly explain anaesthetic and pain management protocols for this particular ACLT surgery.

- Please explain procedure for intraarticular injections in rats. Were these done under general anaesthesia? With which needle size? Volume injected per joint? Vehicle substance? Were cells administered directly after thawing or were cultured for a few days after thawing to readjust conditions? If the former, viability of cells post thawing?

Macroscopic evaluation

- Line 171: even if it might be obvious, please state that at this time point (8 weeks) animals were euthanized as explained before for tissue harvesting.

Statistical analysis

- How was normality assessed, e.g. Shapiro Wilk, Kolmogorov Smirnov...? Since non-parametric tests were used (Kruskall Wallis, Mann-Whitney) I guess that normality was checked and not met. Please clarify.

- In addition, when three groups are compared (e.g. control, HGF-silenced MSCs, control MSCs), the customary strategy is to run a test like Kruskall-Wallis with a posthoc such as Dunns or similar, and not to subsequently conduct pairwise analysis like Mann-Whitney. Please revise and consider amendment.

- Why customary p<0.05, p<0.01 and p<0.001 were not used but instead arbitrary p value levels are considered? It is not technically incorrect, but a bit odd to make this choice.

Results

Hgf knockdown

- Figure 1a does not provide much information. It could be removed or, if kept, it should be enriched with further details like a whole study design representation of the in vitro assays (gene expression, protein secretion, chondrogenesis), the n of animals, etc.

- Figure 1b: the amplification plot is not needed and just the table should be kept. As mentioned in methodology, number of biological and technical replicates should be included. I guess it was only 1 replicate and thus there is no mean or deviation to show in terms of gene expression. In the table, it would be useful to include value 1 for the reference sample, even to represent these numbers (1 and 0.3) in a bar graph for visually representing the downregulation of HGF.

- Figure 1c: Were statistically significant differences found among negative control (whatever it is, w/o cells), control MSCs and HGF-silenced MSCs in terms of HGF secretion? Or along the time i.e. among 1, 3 and 7 days? This should also be explained in the main text.

In vitro analysis

- Were statistically significant differences found between control and silenced-HGF MSCs in terms of weight or diameter of the chondrogeneic pellets produced?

- Number of biological and technical replicates in the chondrogeneic assay?

- Figure 2: Even if quantification of histological parameters in chondrogeneic pellets has not been performed, there are some differences between control and silenced HGF MSCs that can be qualitatively described in the main text, such as different staining intensity of Safranin O, for example.

- Figure 2 caption needs to provide more detail into all the data presented (types of staining and immunostaining).

Discussion

- Lines 414 – 416: “The significant reduction in MSC therapeutic efficacy following Hgf knockdown suggests that this growth factor plays a crucial role in orchestrating the complex interplay of these protective mechanisms”. This is an overstatement. First, even though the score reduction was statistically significant over control, such reduction was not so dramatic. Second, and more importantly, from these results it cannot be concluded which exact mechanisms plays HGF in relation to the multimodal action of MSCs, since other mediators and their interactions have not been assessed (please also see general comments).

- Lines 420 – 424: The choice of the mismatched scenario using two rat strains should be explained in the introduction or in the study design section so the reader can understand the suitability of the mismatched design from the beginning of the manuscript.

- Lines 424 – 425: This sentence should be revised. As already mentioned, the lack of MHC-matched control (i.e. syngenic) does not allow inferring if the cells would have promoted more or less effect depending on the matching degree. Maybe matched cells would have not elicited more therapeutic effects in the conditions of this study, so it cannot be inferred that it is a positive results that mismatched cells still elicited therapeutic effect.

**Do you want your identity to be public for this peer review?** For information about this choice, including consent withdrawal, please see our Privacy Policy

Reviewer #1: No

Reviewer #2: No

---

## [Author Response · Author response to Decision Letter 1]

18 Aug 2025

Review Comments to the Author

Reviewer #1:

The study asks whether major histocompatibility mismatch in synovial-derived MSCs will block the chondroprotective effect previously shown by the same group in a different rat OA induction model.

The study also asks whether the synovial-derived MSC’s ability to express HGF is necessary for their therapeutic effect.

The major findings of the study are that synovial derived MSCs appear to still have a chondroprotective effect despite MHC mismatch; and that this effect was not observed if synovial derived MSCs with siRNA induced HGF knockdown were used.

Overall comments:

The strengths of the study are the well-controlled designs, reproducibility obtained, and careful analyses.

The weaknesses of the study are the lack of translational relevance of the ACLT model as a clinical model of human OA induction in which damage is much more common in the femoral condyle;

We acknowledge that the ACLT model does not fully replicate the spatial distribution of cartilage damage typically observed in human OA, where the femoral condyle is often more severely affected. However, the ACLT model remains a widely accepted and standardized model for post-traumatic osteoarthritis and provides consistent induction of joint instability and inflammation. Importantly, our aim in conducting this study was not to replicate the entire clinical presentation of OA, but to evaluate the relative therapeutic contribution of HGF within synovial MSCs under immunologically challenging conditions using MHC-mismatched transplantation. In this context, the ACLT model offers a reproducible and appropriate platform for assessing therapeutic differences in MSC-based interventions. We have added the following paragraph to the Discussion:

“Several limitations of this study should be noted. One was that we used the anterior cruciate ligament transection (ACLT) model, which, although widely adopted for post-traumatic osteoarthritis research, may not fully recapitulate the complex pathology of human OA. While this model allows for reproducibility and controlled OA induction, its translational relevance remains limited [26].”

the lack of translational relevance of the MSC injection regimen as a clinical injection regimen where repeated injections would not be feasible;

We acknowledge the reviewer's comment regarding the translational limitations of the repeated MSC injection regimen used in this study. While multiple intra-articular injections may not be routinely feasible in clinical settings due to concerns about patient burden and cost, they are commonly employed in preclinical studies to maximize the opportunity for detecting therapeutic effects and to understand dose-response dynamics under controlled conditions.

Our goal in using repeated injections was to ensure sufficient exposure of transplanted MSCs within an immunologically challenging environment (MHC-mismatched model) and to achieve a robust assessment of the contribution of HGF to therapeutic efficacy. This design helped reduce variability and clarify the biological role of HGF in the context of allogeneic MSC therapy. Nonetheless, we agree that single-dose or limited-injection regimens would more closely reflect clinical practice, and we have noted this as a limitation in the revised Discussion section, as follows:

“Another limitation was that we administered synovial MSCs via intra-articular injection once weekly for 7 weeks, which differs from typical clinical protocols where such frequent dosing may not be feasible [4]. Future studies should explore more clinically relevant dosing regimens.”

and lack of consideration of other signals affected by HGF knockdown in the MSCs that may be impairing their activity versus the author’s inference that the chondroprotective activity of the cells is due exclusively to HGF knockdown.

We thank the reviewer for this insightful comment. We fully agree that HGF knockdown may influence more than just HGF secretion and may also affect other signaling pathways within MSCs, potentially contributing to the observed reduction in therapeutic efficacy. Our study was designed as an observational investigation to explore the role of HGF in MSC-mediated chondroprotection under MHC-mismatched conditions rather than as a study that would definitively establish mechanistic causality.

To avoid overinterpretation, we have revised the relevant statements in the Discussion to clarify that while our findings suggest a contribution of HGF to the therapeutic effects, they do not exclude the possibility that additional molecular changes associated with HGF knockdown may also play a role. We have also added this point as a limitation of the study. Based on these reviewer comments, we have substantially revised the Limitations section as follows:

“Our mechanistic analysis focused specifically on HGF function through siRNA knockdown. While this approach demonstrated the contribution of HGF to MSC efficacy, we did not conduct a comprehensive evaluation of whether Hgf knockdown affected other chondroprotective factors, such as PRG4 or BMP2/6 [3], nor did we perform rescue experiments to isolate the specific contribution made by HGF. Future studies incorporating secretome analysis and rescue experiments would provide more definitive mechanistic insights.”

Specific comments:

Describe size of needle used in MSC injections.

We have added the needle size used for MSC injections as follows:

“Under isoflurane anesthesia, a 28-gauge needle was used for intra-articular injection of 50 μL of the following assigned treatments into the knee joints of the animals in each group:”

Describe if the MHC mismatched cell injections led to any immune response in the animal – eg, presence of lymphocytes in the synovium should be examined.

We thank the reviewer for this important comment regarding the assessment of immune responses following MHC-mismatched MSC administration. We agree that evaluating immune cell infiltration, particularly lymphocytes in the synovium, would provide valuable insights into the immunological consequences of allogeneic MSC transplantation.

In response to this comment, we have added a section titled "Knee findings after cell administration" to describe our observations regarding potential immune responses. As detailed in this new section, we assessed immune responses by examining knee swelling and joint tissue hypertrophy during postmortem examination following synovial MSC administration. However, we observed joint swelling and tissue thickening uniformly across all groups, with no specificity to the cell-administered groups, suggesting that these changes were primarily related to the ACLT-induced osteoarthritis rather than to immune rejection of the transplanted cells.

We have added the following paragraph to the Results:

“Knee findings after cell administration

The induction of immune responses following the administration of MHC-mismatched cells was assessed by examining knee swelling and joint tissue hypertrophy during postmortem examination after synovial MSC administration. However, joint swelling and tissue thickening were observed uniformly across all groups and were not specific to the cell-administered group.”

The authors have previously identified Prg4 and BMP2,6 as additional potentially pro-chondrogenic factors expressed by synovial MSCs. Were the levels of Prg4 and BMP2,6 mRNA and protein also decreased in the cells by HGF knockdown? Without this information, it is an overstatement to attribute the chondroprotective.

We thank the reviewer for this important and insightful comment. We fully agree that the potential involvement of other pro-chondrogenic factors, such as Prg4 and BMP2/6, in mediating the chondroprotective effects of synovial MSCs warrants further investigation. However, due to time constraints—specifically, the limited five-week period allocated for the revision—performing additional experiments involving mRNA and protein analysis of these factors was not feasible within the current study timeline. We plan to address this important mechanistic question in a follow-up study, and we have added the following statement acknowledging this limitation to the Discussion section:

“Our mechanistic analysis focused specifically on HGF function through siRNA knockdown. While this approach demonstrated the contribution of HGF to MSC efficacy, we did not conduct a comprehensive evaluation of whether Hgf knockdown affected other chondroprotective factors, such as PRG4 or BMP2/6 [3], nor did we perform rescue experiments to isolate the specific contribution made by HGF. Future studies incorporating secretome analysis and rescue experiments would provide more definitive mechanistic insights.”

Reviewer #2:

This manuscript presents a study aimed at determining the implication of hepatocyte growth factor (HGF) in the therapeutic effects of synovial derived mesenchymal stem cells (MSCs) in a rat osteoarthritis (OA) model. To do this, both in vitro and in vivo studies have been conducted. First, HGF expression is silenced, and it is demonstrated that this silencing has a negative effect for the chondrogenic differentiation of synovial MSCs in vitro. Second, both HGF-silenced and control MSCs are administered into rat OA knees (as well as no-cells control), and the effects of these administrations is measured by macroscopic and histological findings in cartilage. Of note, the in vivo study is designed for the MSC administration to be MHC-mismatched, using two different rat strains as donors (ACI) and recipients (Lewis).

Overall, the manuscript is well written, the goals are clear (but could be more precisely defined), the study design and methods are appropriate (they could be complemented, though), and the results are generally clearly presented. Limitations are discussed and conclusions are based on the results. In general, this is an interesting and formally adequate manuscript, however some points need attention prior to further considering it for publication.

Please see below main general comments, and later more specific comments pointed out along the manuscript. I hope these points can help authors making the most of their manuscript, but if they disagree with any comment/suggestion, please kindly explain why.

General comments:

- I strongly recommend putting more emphasis into the novelty of the study. The introduction is very brief, and, in my opinion, it does not provide enough context to understand the relevance of the present work. Previous studies have already assessed the role of HGF in MSC mechanisms in vitro and in vivo, and some of them have also assessed knockdown cells in different disease models, so the authors should answer very clearly in the introduction: what is the novelty of our study? What does our study adds to the current knowledge?

We sincerely thank the reviewer for this valuable feedback regarding the novelty and context of our study. We completely agree that the previous introduction was insufficient in establishing the unique contribution of our work and its relevance to the current body of knowledge.

In response to this comment, we have substantially expanded and restructured the introduction to better contextualize our study within the existing literature and to more clearly articulate its novelty. The revised introduction now provides a comprehensive background on osteoarthritis pathophysiology, the advantages and challenges of allogeneic MSC therapy, and the specific knowledge gaps that our study addresses.

Specifically, we have clarified that while previous studies have examined HGF's role in MSC mechanisms, our work uniquely focuses on the functional importance of endogenously secreted HGF under fully MHC-mismatched allogeneic conditions—a clinically relevant scenario that remains poorly understood. Unlike previous studies that primarily investigated HGF overexpression or exogenous HGF administration, our approach uses loss-of-function analysis through siRNA knockdown to determine whether endogenous HGF secretion is essential for MSC therapeutic efficacy in an immunologically challenging environment.

Furthermore, we have emphasized that our study addresses critical translational questions, namely, whether the therapeutic benefits of synovial MSCs can be maintained even when donor-recipient immune compatibility is compromised and what role HGF plays in this context. Knowing the answers to these questions is essential for advancing allogeneic MSC therapy toward clinical application, where perfect MHC matching is often impractical.

We have rewritten the introduction as follows:

“Introduction

Osteoarthritis (OA) is a degenerative joint disease characterized by cartilage degradation, synovial inflammation, and structural changes in the subchondral bone that ultimately lead to joint dysfunction and chronic pain [1]. OA progression is driven by multiple factors, including cartilage matrix breakdown, chondrocyte apoptosis, and elevated levels of inflammatory cytokines [2]. Due to this multifactorial pathophysiology, therapeutic approaches that can target several pathways simultaneously—such as cell-based therapies—are attracting growing interest.

Synovial mesenchymal stem cells (MSCs) have shown promising chondroprotective effects when administered intra-articularly, as demonstrated in both preclinical [3] and clinical studies [4]. We previously reported that intra-articular injection of autologous synovial MSCs can suppress OA progression in humans [4]. However, autologous transplantation presents several challenges, including donor site morbidity, extended culture periods, and high manufacturing costs. These limitations have prompted increasing interest in allogeneic MSC therapy, which offers advantages in terms of availability, standardization, and scalability.

A key concern arising during allogeneic MSC therapy is immunological incompatibility between the donor and recipient, particularly regarding major histocompatibility complex (MHC) antigens. Previous studies using rat meniscectomy models have shown that meniscal regeneration is poorer following transplantation of fully MHC-mismatched MSCs than following transplantation of syngeneic or partially matched MSCs [5, 6]. This observation suggests that immune rejection can negatively impact therapeutic outcomes; however, the efficacy of using fully mismatched MSCs in OA models remains poorly understood.

Given the limited understanding of the underlying mechanisms controlling the therapeutic responses observed following the transplantation of MHC-mismatched MSCs in OA, clarifying the roles of paracrine factors secreted by MSCs becomes essential. Among the various paracrine factors, hepatocyte growth factor (HGF) is of particular interest due to its well-documented anti-inflammatory, antifibrotic, and tissue-regenerative properties [7]. HGF stimulates chondrocyte proliferation, enhances matrix production, and suppresses catabolic cytokines, and the benefits of the administration of exogenous HGF or HGF-overexpressing MSCs have been reported [8, 9]. Nevertheless, the functional importance of endogenously secreted HGF—especially under MHC-mismatched allogeneic conditions—remains unclear. However, the possibility remains that HGF may serve as a critical mediator of MSC therapeutic efficacy, even in immunologically challenging environments.

The aim of the present study was to clarify the functional role of HGF in establishing the chondroprotective effects of synovial MSCs under fully MHC-mismatched allogeneic conditions. We hypothesized that HGF serves as a key mediator of MSC efficacy, even under immunologically challenging conditions. To test this hypothesis, we employed a rat OA model using ACI rats (donors) and Lewis rats (recipients), which differ markedly in MHC class I and II antigens and are known to elicit robust immune responses [5]. Synovial MSCs were transfected with Hgf siRNA and injected intra-articularly, and the effects of HGF suppression on both in vivo chondroprotective efficacy and in vitro chondrogenic capacity were evaluated. Importantly, our study specifically focuses on OA cartilage rather than meniscal tissue, and employs HGF knockdown to mechanistically

---

## [Decision Letter · Decision Letter 1]

3 Sep 2025

Dear Dr. Sekiya,

Thank you for submitting your manuscript to PLOS ONE. After careful consideration, we feel that it has merit but does not fully meet PLOS ONE’s publication criteria as it currently stands. Therefore, we invite you to submit a revised version of the manuscript that addresses the points raised during the review process.

We look forward to receiving your revised manuscript.

Kind regards,

Xindie Zhou

Academic Editor

PLOS ONE

Journal Requirements:

Reviewers' comments:

Reviewer's Responses to Questions

**Comments to the Author**

Reviewer #2: (No Response)

2. Is the manuscript technically sound, and do the data support the conclusions?

Reviewer #2: Yes

3. Has the statistical analysis been performed appropriately and rigorously?

Reviewer #2: Yes

4. Have the authors made all data underlying the findings in their manuscript fully available?

Reviewer #2: Yes

5. Is the manuscript presented in an intelligible fashion and written in standard English?

Reviewer #2: Yes

Reviewer #2: Authors have done a remarkable job revising the manuscript, which has significantly improved. It is particularly worth of mentioning the great work with the Introduction, which is convincing yet concise and clearly highlighting the main points. I consider that the manuscript is in good shape for acceptance, yet a few minor points need attention:

- Regarding the procedure for MSC injection, the vehicle substance is not yet disclosed. May it be CP-1 High Grade cryoprotectant used for freezing the cells? If so, please clearly state.

- Regarding the sex of the animals of the study: I understand that it is a common approach to use males as donors and females as recipients in order to track the cells using Y chromosome markers, and I think this can be a valid point when designing the study, even though tracking of male donor cells was not pursued. I also understand that including a gender dimension in the in vivo studies can be challenging and require extra resources. However, I feel that the statement “we believe the sex difference did not influence thetherapeutic outcomes observed” (lines 559-560 of the Discussion) is not supported by evidence (e.g. doi: 10.21037/atm-23-1546) and thus should be removed.

- In line 480 of the Discussion, the statement “even in the presence of strong allogeneic immune responses” is an assumption, since the immune response against the MSCs was not evaluated. Please consider removing this sentence to avoid misleading information.

**Do you want your identity to be public for this peer review?** For information about this choice, including consent withdrawal, please see our Privacy Policy

Reviewer #2: No

---

## [Author Response · Author response to Decision Letter 2]

4 Sep 2025

Review Comments to the Author

Reviewer #2:

Authors have done a remarkable job revising the manuscript, which has significantly improved. It is particularly worth of mentioning the great work with the Introduction, which is convincing yet concise and clearly highlighting the main points. I consider that the manuscript is in good shape for acceptance, yet a few minor points need attention:

- Regarding the procedure for MSC injection, the vehicle substance is not yet disclosed. May it be CP-1 High Grade cryoprotectant used for freezing the cells? If so, please clearly state.

We appreciate the reviewer’s comment and have revised the Methods section to explicitly state the use of CP-1 High Grade cryoprotectant as the vehicle for MSC cryopreservation and injection. The relevant passages now read as follows:

“Isolation of rat synovial MSC

Rat synovial MSCs were isolated as previously described [10]. After an adaptation period of at least 7 days, male ACI rats were euthanized by exsanguination from the inferior vena cava under isoflurane anesthesia. Synovial tissue harvested from the infrapatellar fat pad of both knees from 15 rats was pooled and minced, followed by digestion with Liberase MNP-S (Roche Diagnostics Corp., IN, USA) in a water bath at 37°C for 2 h. The resulting isolated cells were cultured in α-minimum essential medium (αMEM; FUJIFILM Wako Pure Chemical Corp., Osaka, Japan) supplemented with 20% fetal bovine serum (FBS; Thermo Fisher Scientific, MA, USA) for 8 days at 37°C in 5% CO2. The cells were then harvested and cryopreserved as original stocks in CP-1 High Grade cryoprotectant (Kyokuto Pharmaceutical Industrial Co. Ltd., Tokyo, Japan) at 150°C (CLN-1700CWE, Nihon Freezer, Tokyo, Japan) (Passage 0). These isolation procedures were performed in three separate sessions. For experiments, synovial MSCs from the original stocks (Passage 0) were cultured for 7 days, harvested, and cryopreserved in CP-1 High Grade cryoprotectant (Passage 1). The cells were used for subsequent analyses without any sorting [6]. For cryopreservation, CP-1 High Grade was supplemented with 25% human recombinant albumin expressed in plants (Albumin, Human, recombinant expressed in plants; FUJIFILM Wako Pure Chemical Corp.), resulting in a final composition of 6% hydroxyethyl starch (HES), 5% dimethyl sulfoxide (DMSO), and 4% human recombinant albumin.”

“One week after surgery, the rats were randomly divided into three groups of 10 animals each (n = 10), resulting in 20 knee joints per group (n = 20). Under isoflurane anesthesia, a 28-gauge needle was used for intra-articular injection of 50 μL of the following assigned treatments into the knee joints of the animals in each group:

1. Hgf siRNA-transfected MSCs (1 × 10⁶ cells),

2. Negative control siRNA-transfected MSCs (1 × 10⁶ cells), or

3. Vehicle alone (CP-1 High Grade cryoprotectant without MSCs).”

- Regarding the sex of the animals of the study: I understand that it is a common approach to use males as donors and females as recipients in order to track the cells using Y chromosome markers, and I think this can be a valid point when designing the study, even though tracking of male donor cells was not pursued. I also understand that including a gender dimension in the in vivo studies can be challenging and require extra resources. However, I feel that the statement “we believe the sex difference did not influence the therapeutic outcomes observed” (lines 559-560 of the Discussion) is not supported by evidence (e.g. doi: 10.21037/atm-23-1546) and thus should be removed.

We acknowledge the reviewer’s important observation regarding our statement on sex differences in therapeutic outcomes. As the reviewer correctly points out, our assertion that "we believe the sex difference did not influence the therapeutic outcomes observed" (lines 559–560) was not supported by direct evidence, since our study was not designed to systematically investigate sex-based differences in treatment response. Recognizing the growing body of literature emphasizing the influence of sex on therapeutic outcomes, and given that we did not collect data to address this question, we agree with the reviewer's concern and have therefore removed this unsupported statement from the manuscript. We have rewritten the paragraph as follows:

“The present analysis did not include sex-based cell tracking, although male donor rats and female recipient rats have been used in the past to enable the potential tracking of donor cells using Y chromosome-specific markers [29]; However, our aim was not to evaluate sex-specific effects, and this limitation should be considered when interpreting the results.”

- In line 480 of the Discussion, the statement “even in the presence of strong allogeneic immune responses” is an assumption, since the immune response against the MSCs was not evaluated. Please consider removing this sentence to avoid misleading information.

We are grateful to the reviewer for highlighting this important issue. The reviewer is correct that our statement “even in the presence of strong allogeneic immune responses” (line 480) was an assumption rather than a conclusion supported by data, as our study design did not directly evaluate immune responses against the transplanted MSCs. Recognizing that such assumptions without proper validation may mislead readers, we agree with the reviewer’s suggestion and have removed this statement from the Discussion to maintain accuracy and scientific rigor.

---

## [Decision Letter · Decision Letter 2]

10 Sep 2025

MHC-mismatched synovial mesenchymal stem cell injections delay knee osteoarthritis progression through hepatocyte growth factor secretion in rats

PONE-D-25-18549R2

Dear Dr. Sekiya,

We’re pleased to inform you that your manuscript has been judged scientifically suitable for publication and will be formally accepted for publication once it meets all outstanding technical requirements.

Kind regards,

Xindie Zhou

Academic Editor

PLOS ONE

Additional Editor Comments (optional):

Reviewer #2:

Reviewers' comments:

Reviewer's Responses to Questions

**Comments to the Author**

Reviewer #2: All comments have been addressed

2. Is the manuscript technically sound, and do the data support the conclusions?

Reviewer #2: Yes

3. Has the statistical analysis been performed appropriately and rigorously?

Reviewer #2: Yes

4. Have the authors made all data underlying the findings in their manuscript fully available?

Reviewer #2: Yes

5. Is the manuscript presented in an intelligible fashion and written in standard English?

Reviewer #2: Yes

Reviewer #2: Authors have appropriately addressed the last few pending points. I have no further comments and consider the manuscript suitable for publication.

**Do you want your identity to be public for this peer review?** For information about this choice, including consent withdrawal, please see our Privacy Policy

Reviewer #2: No

---

## [Editor Report · Acceptance letter]

PONE-D-25-18549R2

PLOS ONE

Dear Dr. Sekiya,

I'm pleased to inform you that your manuscript has been deemed suitable for publication in PLOS ONE. Congratulations! Your manuscript is now being handed over to our production team.

Kind regards,

on behalf of

Dr. Xindie Zhou

Academic Editor

PLOS ONE